Multi-scale immunoepidemiological modeling of within-host and between-host HIV dynamics: systematic review of mathematical models

Dorratoltaj Nargesalsadat 1
Nikin-Beers Ryan 2
Ciupe Stanca M. 2
Eubank Stephen G. 3
Abbas Kaja M. kaja.abbas@vt.edu 1
1 Department of Population Health Sciences, Virginia Tech , Blacksburg , United States of America
2 Department of Mathematics, Virginia Tech , Blacksburg , United States of America
3 Biocomplexity Institute, Virginia Tech , Blacksburg , United States of America
Palazón-Bru Antonio
Electronic publication date: 2017 Sep 28
Publication date: 2017
Volume: 5
Electronic Location ID: e3877
Received 2017 Jul 6; Accepted 2017 Sep 11
Copyright: ©2017 Dorratoltaj et al.
Copyright year: 2017
Copyright holder: Dorratoltaj et al.
License: This is an open access article distributed under the terms of the Creative Commons Attribution License, which permits unrestricted use, distribution, reproduction and adaptation in any medium and for any purpose provided that it is properly attributed. For attribution, the original author(s), title, publication source (PeerJ) and either DOI or URL of the article must be cited.
License URL: https://creativecommons.org/licenses/by/4.0/

Keywords: Super infection, Co-infection, HIV acquisition and transmission, Evolution, Immunoepidemiology, Multi-scale model, Immune-viral dynamics, HIV, Drug resistance, Super-infection

Funding: The authors received no funding for this work.

==============================
Objective

The objective of this study is to conduct a systematic review of multi-scale HIV immunoepidemiological models to improve our understanding of the synergistic impact between the HIV viral-immune dynamics at the individual level and HIV transmission dynamics at the population level.

Background

While within-host and between-host models of HIV dynamics have been well studied at a single scale, connecting the immunological and epidemiological scales through multi-scale models is an emerging method to infer the synergistic dynamics of HIV at the individual and population levels.

Methods

We reviewed nine articles using the PRISMA (Preferred Reporting Items for Systematic Reviews and Meta-Analyses) framework that focused on the synergistic dynamics of HIV immunoepidemiological models at the individual and population levels.

Results

HIV immunoepidemiological models simulate viral immune dynamics at the within-host scale and the epidemiological transmission dynamics at the between-host scale. They account for longitudinal changes in the immune viral dynamics of HIV+ individuals, and their corresponding impact on the transmission dynamics in the population. They are useful to analyze the dynamics of HIV super-infection, co-infection, drug resistance, evolution, and treatment in HIV+ individuals, and their impact on the epidemic pathways in the population. We illustrate the coupling mechanisms of the within-host and between-host scales, their mathematical implementation, and the clinical and public health problems that are appropriate for analysis using HIV immunoepidemiological models.

Conclusion

HIV immunoepidemiological models connect the within-host immune dynamics at the individual level and the epidemiological transmission dynamics at the population level. While multi-scale models add complexity over a single-scale model, they account for the time varying immune viral response of HIV+ individuals, and the corresponding impact on the time-varying risk of transmission of HIV+ individuals to other susceptibles in the population.

Introduction

HIV prevalence and mortality were 38.8 million and 1.2 million deaths respectively in 2015, with annual incidence being relatively constant at 2.6 million per year from 2005 to 2015 (Wang et al., 2016). Access to big data and emergence of unanswered questions enable novel methods of mathematical models to connect within-host immune viral dynamics at the individual level, and the between-host epidemiological transmission of infectious diseases at the population level (Gog et al., 2015). Mathematical models of HIV dynamics have been extensively studied using single-scale based models at the immunological and epidemiological scales (Perelson & Ribeiro, 2013; Akpa & Oyejola, 2010). The immunological models focus on the within-host immune viral dynamics at the individual level, while the epidemiological models focus on the between-host transmission dynamics at the population level. Multi-scale immunoepidemiological modeling is an emerging method to study the synergistic dynamics of HIV at the individual and population levels (DebRoy & Martcheva, 2008; Yeghiazarian, Cumberland & Yang, 2013; Handel & Rohani, 2015).

Epidemiological models

Epidemiological modeling of HIV infection started in 1985 (Curran et al., 1985). Epidemiological models of HIV infections assign each individual to one of the following states: susceptible or infected. Infected individuals may transmit HIV to susceptible hosts with the same transmission rate over the course of disease, and experience specific duration of infection (Isham, 1988; Hyman & Ann Stanley, 1988; Haberman, 1990). However, time since infection, other co-infections, and a host’s biological factors such as age, sex, genetic susceptibility, and immune status cause variation in infectiousness of HIV+ individuals (Cassels, Clark & Morris, 2008). Host heterogeneity among different ages, gender and risk groups is significant due to the multiple routes of transmission—sexual transmission, intravenous transmission through needle sharing, blood transfusion, and mother-to-child vertical transmission.

Immunological models

Within-host models of HIV at the individual level study the dynamics of HIV and target immune cells—CD4+ T cells, macrophages, and dendritic cells. The complexity of the models vary from molecular level (Reddy & Yin, 1999; Zarrabi et al., 2010; Hosseini & Mac Gabhann, 2012), cellular level (Anderson & May, 1992; McLean, 1993; Ho et al., 1995; Perelson et al., 1996; Kirschner, 1996; De Boer & Perelson, 1998; Banks et al., 2008; Hosseini & Mac Gabhann, 2012; Perelson & Ribeiro, 2013), and tissue level (Spouge, Shrager & Dimitrov, 1996). The within-host immunological models analyze the mechanisms of HIV pathogenesis and prognosis from acute, latent and late stages of HIV infection to AIDS phase.

Immunoepidemiological models

Figure 1 illustrates that the transmission dynamics of HIV in the population is dependent on the immune viral dynamics of HIV+ individuals. Immunoepidemiological models factor the HIV transmission dynamics at the population level as a function of within-host immune viral responses at the individual level (DebRoy & Martcheva, 2008; Yeghiazarian, Cumberland & Yang, 2013; Hellriegel, 2001).

Figure 1 Within-host immune-viral dynamics and between-host transmission dynamics of HIV.

HIV spreads in the population from infected individuals to susceptibles through sexual contact, intravenous drug use, blood transfusion and mother-to-child vertical transmission. HIV immune-viral dynamics determine the time-varying viral load within each infected individual.

Clinical and public health significance

HIV immunoepidemiological models focus on solutions for the following questions of clinical and public health significance (Feng et al., 2011):

• How does within-host immune-viral dynamics of HIV affect incidence at the population level?

• How does population level transmission dynamics of HIV affect viral evolution at the individual level?

In this study, we review the multi-scale modeling methods that connect the within-host and between-host scales of HIV models. Understanding the relation between these two scales is key to understand HIV prognosis, transmission risk, and intervention effectiveness (Pepin et al., 2010).

Methods

Search strategy

We searched the PubMed database for articles published from December 1, 1985 to June 1, 2017 with the terms: (HIV and (“multi-scale” or “immunoepidemiology” or “nested model” or (“within-host” and “between host”) or (“within-host” and “among host”) or (“within-host” and (“epidemiology” or “epidemiological”)))).

Data abstraction and synthesis

The data abstraction and synthesis process was conducted by two authors (ND and RNB) independently, and includes the following four steps: identification, screening, eligibility, and inclusion. We resolved discordant decisions through consensus. During the identification step, articles were identified using the above search strategy. During the screening step, duplicate articles were removed, and titles and abstract of the remaining articles were screened to determine their relevance to our study. During the eligibility step, full texts of the articles were analyzed to determine their relevance to our study.

Inclusion and exclusion criteria

The inclusion criteria were articles focused on multi-scale immunoepidemiological modeling of HIV dynamics. The exclusion criteria were articles that focused on genetic epidemiology, molecular epidemiology, parasitology, ecology, evolutionary study, and experimental studies.

PRISMA process

Figure 2 illustrates the process flow diagram of identification, screening, eligibility, and inclusion of articles for the systematic review, using the PRISMA (Preferred Reporting Items for Systematic Reviews and Meta-Analyses) framework (Moher et al., 2009). 89 articles were uniquely identified, 66 articles were screened out, and nine articles were found eligible to be included in this systematic review. This systematic review includes a qualitative synthesis and does not include the quantitative synthesis of a meta-analysis (not applicable for this study).

Figure 2 PRISMA flow-diagram.

PRISMA (Preferred Reporting Items for Systematic Reviews and Meta-Analyses) flow-diagram of articles’ identification, screening, eligibility and inclusion in the systematic review. A total of nine studies are included in this systematic review of multi-scale immunoepidemiological modeling of within-host and between-host HIV dynamics.

Results

Table 1 illustrates the characteristics of HIV immunoepidemiological modeling studies included in this systematic review. The objective, model implementation, immunoepidemiological link between within-host and between-host models, and significant inferences of these studies are summarized in the table.

Table 1 Characteristics of HIV immunoepidemiological modeling studies.

The study topic, objective, model implementation, immunoepidemiological link between within-host and between-host models, and inferences of the studies included in the systematic review are summarized.

Study	Topic	Objective	Implementation	Immunoepidemiological link	Inferences	
Martcheva & Li (2013)	Super-infection	How does HIV super-infection affect population dynamics?	Partial differential equations	Transmission rate between hosts and death rate of individuals depend on viral load within host over time.	In certain cases, decreasing viral load can cause higher prevalence of HIV since infected individuals may live longer; oscillations at population level do not occur in superinfection, contrasting previous studies that did not use linked models.	
Saenz & Bonhoeffer (2013)	Drug resistance	How do the dynamics of drug-sensitive and drug-resistant HIV strains within hosts affect the prevalence of drug-resistant strains in the population?	Partial differential equations	Transmission rate between hosts depends on viral load within host over time.	Increasing early initiation and coverage decreases total prevalence upto an optimal treatment coverage level but increases incidence and prevalence of drug resistant infections; above the optimal treatment coverage level, number of infections may not decrease in the long term and can even increase.	
Lythgoe, Pellis & Fraser (2013)	Evolution	How does competition between strains within-host affect evolution of HIV virulence?	Integro-differential equations with delay	Strain-specific infectivity rate between hosts depends on frequency of strains within-host.	Small rates of within-host evolution modestly increase HIV virulence while maximizing transmission potential; high rates of within-host evolution largely increase HIV virulence but lower transmission potential.	
Doekes, Fraser & Lythgoe (2017)	Evolution	How does latent reservoir of infected CD4+ T cells affect the types of strains of HIV that will evolve within and between hosts?	Integro-differential equations with delay	Strain-specific infectivity rate between hosts depends on frequency of strain in actively infected CD4+ T cells within-host.	Relatively large latent reservoirs cause delay to within-host evolutionary processes, which select for moderately virulent strains that optimize transmission at the population level; with no reservoir, highly virulent strains are selected for within-host that do not optimize transmission at the population level.	
Cuadros & García-Ramos (2012)	Co-infection	How does co-infection affect the HIV replication capacity?	Ordinary differential equations	Transmission rate between hosts depends on steady-state of viral load within host.	Impact of co-infection increases as average set-point viral load of population increases.	
Yeghiazarian, Cumberland & Yang (2013)	ART	How does the timing of antiretroviral therapy (ART) in individuals affect the spread of HIV?	Individual-based model	Transmission rate to each susceptible partner depends on viral load of infected individual.	Beginning ART during acute infection is most effective for reducing spread of HIV.	
Shen, Xiao & Rong (2015)	ART	How does antiretroviral therapy (ART) affect HIV prevalence?	Partial differential equations	Transmission rate depends on saturated viral load within-host, and varies between stages of infection.	While ART decreases the viral load and infectiousness of each infected host, in certain cases, this can lead to higher spread of HIV throughout the population because these infected individuals live longer; HIV can still be controlled in these cases if drug effectiveness is high.	
Sun et al. (2016)	ART	How does antiretroviral therapy (ART) affect HIV prevalence?	Individual-based model	Transmission rate to each susceptible partner depends on viral load of infected individual.	Initiating ART early causes lower transmission of HIV in population; however, when ART efficacy decreases with emergence of drug resistance, early treatment leads to higher HIV spread in the population because the prevalence of drug resistant strains increases rapidly.	
Metzger, Lloyd-Smith & Weinberger (2011)	TIPs	How does introduction of therapeutic interfering particles (TIPs) affect HIV prevalence?	Ordinary differential equations	Transmission rate between hosts depends on steady-states of TIP and HIV viral loads within-host.	Deploying TIPs in even small numbers of infected individuals reduces the prevalence of HIV to low levels due to TIPs’ ability to transmit between hosts and target high-risk groups; using TIPs reduces challenges of antiretroviral therapy and vaccines, and complements them.	

Within-host scale of HIV immunoepidemiological models

The within-host scale of HIV immunoepidemiological models simulate the immune-viral dynamics of HIV, which can later be used to determine the impact on transmission between hosts. We categorize the within-host models by whether they model a single strain of HIV, super-infection, drug resistance, evolution, co-infection and therapeutic interfering particles. The immunological scale includes the primary state variables of uninfected CD4+ T cells concentration (T), infected CD4+ T cells concentration (T∗) and viral load (V), and the corresponding parameters for the immune-viral dynamics between these state variables (Anderson & May, 1992; Perelson et al., 1996; De Boer & Perelson, 1998).

HIV infection with single strain

In this approach, it is assumed that there is only one strain of HIV that infects the target cells. No additional features such as mutation, super-infection, or co-infection are considered at the within-host scale. We found three models that include only one strain of HIV at the within-host scale (Shen, Xiao & Rong, 2015; Sun et al., 2016; Yeghiazarian, Cumberland & Yang, 2013). An example of the basic dynamics are shown in Table 2, which also assumes that viral shedding rate (s) has negative effect on the viral load (V) within-host (DebRoy & Martcheva, 2008). This model can be modified to include the effects of drug therapy, which affect the viral production rate and the viral infectivity rate (Shen, Xiao & Rong, 2015; Sun et al., 2016; Yeghiazarian, Cumberland & Yang, 2013).

Table 2 HIV infection with single strain.

Within-host layer of HIV multi-scale model with assumption of single strain HIV infection. The uninfected CD4+ T cells get infected by the free virions and produce HIV virus. CD4+ T cells have the constant reproduction and death rates. HIV induces death rate of infected cells. HIV population increases by production of virus by infected cells, and decreases because of the virus clearance and shedding rate.

Model diagram	
	
Equations	
dTdτ=λ−kTV−dT	
dT∗dτ=kTV−μ+dT∗	
dVdτ=Nμ+dT∗−c+sV−kTV	
Parameters	
λ	Reproduction rate of uninfected cells	
k	Infection rate of uninfected cells	
d	Natural death rate of uninfected cells	
μ	HIV induced death rate of infected cells	
N	HIV production by infected cells	
s	Shedding rate of virus	
c	HIV clearance rate	

HIV super-infection

HIV super-infection occurs when individuals infected with a single HIV strain are infected with a second HIV strain. Martcheva and Li included HIV infection with multiple strains in their model, with the assumption of complete competitive exclusion between the strains at the within-host scale. In this context, the strain with the larger reproduction rate becomes dominant. They studied the impact of virulence of different strains on the equilibrium at the individual and population scales (Martcheva & Li, 2013). Table 3 shows the schematic and formulation of this model.

Table 3 HIV super-infection.

The within-host layer of HIV multi-scale model illustrates the impact of infection with multiple strains of HIV. This model includes the uninfected, infected target CD4+ T cells with different strains, and different strains of free HIV virions. An individual may get infected with drug-resistant and/or drug-susceptible strains. Also, mutations may happen within-host leading to emergence of drug-resistant strains.

Model diagram	
	
Equations	
dTdτ=λ−kiTVi−dT	
dTidτ=kiTVi−μi+dTi	
dVidτ=Niμi+dTi−c+siVi−kiTVi	
Parameters	
λ	Reproduction rate of uninfected cells	
ki	Infection rate of uninfected cells by virus strain i	
μi	HIV induced death rate of infected cell with strain i	
Ni	HIV Production of virus i by infected cells	
si	Shedding rate of virus strain i	
d	Natural death rate of uninfected cells	
c	HIV clearance rate	

HIV drug resistance

Drug resistance can be acquired through mutations of drug-sensitive strains within-host or through direct transmission of drug-resistant strains. Saenz and Bonhoeffer included HIV infection with drug resistant strains in their model, and studied the effects of antiretroviral treatment (ART) on both drug-sensitive and drug-resistant strains (Saenz & Bonhoeffer, 2013). Table 4 shows the schematic and formulation of this model.

Table 4 HIV drug resistance.

The within-host layer of HIV multi-scale model illustrates the uninfected and infected target CD4+ T cells, including drug-sensitive and drug-resistant strains. Mutations from drug-sensitive to drug-resistant or drug-resistant to drug-sensitive strains are studied in this model, and the impact of treatment is also included.

Model diagram	
	
Equations	
dTdτ=λ−1−ϵrtksTVs−1−prtϵrtkrTVr−dT	
dTsdτ=1−msr1−ϵrtksTVs+mrs1−prtϵrtkrTVr−μ+dTs	
dTrdτ=msr1−ϵrtksTVs+1−mrs1−prtϵrtkrTVr−μ+dTr	
dVsdτ=1−ϵpiNsμ+dTs−cVs	
dVrdτ=1−ppiϵpiNrμ+dTr−cVr	
Parameters	
λ	Reproduction rate of uninfected cells	
ks	Infection rate of uninfected cells by drug-sensitive strain	
kr	Infection rate of uninfected cells by drug-resistant strain	
d	Natural death rate of uninfected cells	
μ	HIV induced death rate of infected cells	
c	HIV clearance rate	
ϵrt	Efficacy of reverse transcriptase inhibitor treatment	
ϵpi	Efficacy of protease inhibitor treatment	
Vs	Drug sensitive strain of HIV	
Vr	Drug resistant strain of HIV	
msr	A proportion of infected cells with drug-sensitive strain that produce drug resistant virions	
mrs	A proportion of infected cell with drug-resistant strain that produce drug sensitive virions	
prt	Relative rate of reverse transcriptase inhibitor efficacy for drug resistant strain	
ppi	Relative rate of protease inhibitor efficacy for drug resistant strain	
Ns	Reproduction of HIV virus by drug-sensitive strain	
Nr	Reproduction of HIV virus by drug-resistant strain	

HIV evolution

Studies have modeled HIV viral evolution within-host and its impact on transmission between hosts (Lythgoe, Pellis & Fraser, 2013; Doekes, Fraser & Lythgoe, 2017). They investigate the trade-off between increased virus replication and virulence and decrease in virus transmission. Doekes, Fraser & Lythgoe (2017) also included long-lived reservoirs of latently infected CD4+ T cells to determine their impact on HIV within-host competition.

HIV co-infection

HIV co-infection with sexually transmitted infections among high risk groups (Abu-Raddad et al., 2008), and/or co-infection with endemic infections such as malaria (Cuadros et al., 2011) have direct impact on increasing the transmission rate of both infections. Cuadros and García-Ramos incorporated HIV co-infection dynamics in the within-host immune model (Callaway & Perelson, 2002; Stafford et al., 2000; Nowak & May, 2000) to address increased immune response and increased risk of transmission, and evaluated their impact on HIV epidemics (Cuadros & García-Ramos, 2012). Table 5 shows the schematic and formulation of this model.

Table 5 HIV co-infection.

The within-host layer of HIV multi-scale model illustrates the impact of co-infection. This model includes the uninfected and infected target CD4+ T cells, and free virions. Co-infection increases immune response and the infection rate of immune cells. Therefore, the set-point viral load is higher compared to the case of no co-infection.

Model diagram	
	
Equations	
dTdτ=λ−kTV−dT+1−Tm+T+T∗TmaxrTT	
dT∗dτ=kTV−μ+dT∗+kTmVTm	
dVdτ=Nμ+dT∗−cV	
Parameters	
λ	Reproduction rate of uninfected cells	
k	Infection rate of uninfected cells	
d	Natural death rate of uninfected cells	
μ	HIV induced death rate of infected cells	
N	HIV production by infected cells	
c	HIV clearance rate	
Tm	Activated immune cells against co-infection	
Tmax	Maximum number of immune cells	
rT	Growth rate of non-specific immune cells	
kTm	Infection rate of co-infection	

HIV and therapeutic interfering particles

Therapeutic interfering particles (TIPs) are an emerging drug therapy where therapeutic versions of the pathogen are manufactured to attack viral replication processes and can be transmitted between hosts (Metzger, Lloyd-Smith & Weinberger, 2011). In the within-host model developed by Metzger et al. HIV and TIPs are treated as separate viral strains. The model includes CD4+ T cells infected with HIV only, CD4+ T cells infected with TIPs only, and CD4+ T cells dually infected with HIV and TIPs (Metzger, Lloyd-Smith & Weinberger, 2011).

Between-host scale of HIV immunoepidemiological models

Between-host scales of HIV immunoepidemiological models are based on the susceptible-infectious (SI) epidemic model, which have been used extensively to study HIV transmission dynamics in a homogeneous population and random mixing of susceptibles (S) and HIV+ individuals (I) (Isham, 1988). Table 6 shows the schematic and formulation of the SI epidemic model. Studies have extended the homogeneous population structure of the SI model to incorporate different populations of infected individuals. We categorize the studies by how they divide the infected population, and thus how the transmission rates between these classes differ. We find heterogeneity in HIV transmission rates depending on the stages of HIV infection (Cuadros & García-Ramos, 2012; Yeghiazarian, Cumberland & Yang, 2013; Sun et al., 2016; Shen, Xiao & Rong, 2015), and the dynamics of super-infection (Martcheva & Li, 2013), drug resistance (Saenz & Bonhoeffer, 2013), evolution (Lythgoe, Pellis & Fraser, 2013; Doekes, Fraser & Lythgoe, 2017), and therapeutic interfering particles (Metzger, Lloyd-Smith & Weinberger, 2011).

Table 6 Susceptible-Infected (SI) epidemic model.

The between-host layer of HIV multi-scale model illustrates the random mixing of susceptibles and infected individuals. Susceptibles get infected by the infected individuals. HIV transmission rate (β) depends on the HIV viral load at the within-host scale.

Model diagram	
	
Equations	
dSdt=b−βSI−δS	
dIdt=βSI−α+δI	
Parameters	
S	Number of individuals in the susceptible class	
I	Number of individuals in the infected class	
b	Natural birth rate in the population	
β	HIV transmission rate in the population	
α	Disease induced mortality rate	
δ	Natural death rate in the population	

Acute, latent and late stages of HIV infection

Previous studies have shown that transmission rates differ depending on whether the infected population is in the acute, latent, or AIDS stages (Hollingsworth, Anderson & Fraser, 2008). This conclusion can be incorporated into immunoepidemiological models by categorizing the infected population into different stages (Cuadros & García-Ramos, 2012; Yeghiazarian, Cumberland & Yang, 2013; Sun et al., 2016; Shen, Xiao & Rong, 2015; Saenz & Bonhoeffer, 2013). Cuadros and García-Ramos extended the model so that the HIV+ sub-populations also differed by sexual-risk activity (Cuadros & García-Ramos, 2012). Yeghiazarian et al. divided the infected population into stages to evaluate the timing of treatment initiation at the individual level, and its impact on HIV transmission at the population level. They assumed treatment initiation can start during any stage of HIV infection after diagnosis (Yeghiazarian, Cumberland & Yang, 2013).

HIV super-infection

HIV infected individuals are categorized based on the strains of infection. Due to the assumption of competitive exclusion at the within-host level in the model developed by Martcheva and Li, susceptible individuals only become infected with one of the strains. Thus, only infected individuals having the dominant within-host strain can super-infect individuals with the lesser within-host strain (Martcheva & Li, 2013).

HIV drug resistance

Drug-resistant strains can emerge during antiretroviral therapy (ART) (Rong, Feng & Perelson, 2007), or can be transmitted between individuals who have never been exposed to ART (Hué et al., 2009), which may lead to treatment failure if ART is begun (Hamers et al., 2011). Saenz and Bonhoeffer thus categorize the infected population into those with only drug-sensitive or only drug-resistant strains with or without treatment, and those with drug-sensitive strains that develop drug-resistance while receiving treatment (Saenz & Bonhoeffer, 2013). Table 7 shows the schematic and formulation of this model.

Table 7 HIV drug resistance and treatment impact.

HIV transmission dynamics between drug-sensitive and drug-resistant infected individuals are illustrated. Infected individuals may get infected by the drug-sensitive or drug-resistant strains. A proportion p of infected individuals get treatment, and among the infected individuals with drug-sensitive strains, a proportion q of them develop drug resistance.

Model diagram	
	
Equations	
dSdt=b−βDRSIDR−βDSSIDS−δS	
dIDR0dt=1−pβDRSIDR−α+δIDR0	
dIDRTdt=pβDRSIDR−α+δIDRT	
dIDS0dt=1−pβDSSIDS−α+δIDS0	
dIDSTdt=p1−qβDSSIDS−α+δIDST	
dIDSTrdt=pqβDSSIDS−α+δIDSTr	
Parameters	
b	Natural birth rate in the population	
IDR0	Number of individuals infected with drug-resistant strain and do not receive treatment	
IDRT	Number of individuals infected with drug-resistant strain and receive treatment	
IDS0	Number of individuals infected with drug-sensitive strain and do not receive treatment	
IDST	Number of individuals infected with drug-sensitive strain and receive treatment	
IDSTr	Number of individuals infected with drug-sensitive strain, receive treatment, and develop resistance	
βDR	Drug-resistant HIV transmission rate in the population	
βDS	Drug-sensitive HIV transmission rate in the population	
α	HIV induced mortality rate	
δ	Natural death rate in the population	
p	Proportion of infected individuals who receive treatment	
q	Proportion of infected individuals who receive treatment and develop resistance	
IDR	IDR0 + IDRT	
IDS	IDS0 + IDST + IDSTr	

HIV evolution

Depending on virulence of the strain, infected individuals are categorized by the strain with which they initially became infected (Doekes, Fraser & Lythgoe, 2017; Lythgoe, Pellis & Fraser, 2013). Because it is assumed that all other strains develop from an initial strain and only the most virulent strain is transmitted, infected individuals can end up infecting others with a different strain than they were initially infected. Table 8 shows the schematic and formulation of this model.

HIV and therapeutic interfering particles

The infected population is divided into classes of those infected with HIV only, those infected with Therapeutic Interfering Particles (TIPs) only, and those infected dually with HIV and TIPs. The infected population is also divided into these classes during different stages of infection (Metzger, Lloyd-Smith & Weinberger, 2011). Table 9 shows the schematic and formulation of this model.

Table 8 HIV evolution.

HIV transmission dynamics between infected individuals with different strains are illustrated. Infected individuals with strains i may get infected with another strain j and transmit the dominant strain of HIV.

Model diagram	
	
Equations	
St=b−∑i=1n ∫0TiHit−τe−δτdτ	
Iit= ∫0TiHit−τe−δτdτ−α+δIi	
Hit=StNt∑j=1n ∫0TiβijτHjt−τe−δτdτ	
Parameters	
b	Natural birth rate in the population	
Ti	Time of death after initiation of infection	
Hi	The rate at which new type-i infection occurs	
δ	Natural mortality rate	
Ii	Number of individuals infected with strain i	
βij	Infectivity of strain i in a host originally infected with strain j at time τ since infection.	

Table 9 HIV and therapeutic interfering particles (TIPs).

HIV transmission dynamics between infected individuals with wild type of HIV and TIPs are illustrated. Individuals can get infected with wild type of HIV, TIPS, or both. Infected individuals can get reinfected with both types.

Model diagram	
	
Equations	
dSdt=b−βWISI−βWId1−βTIdSId−βWIdβTIdSId−βTId1−βWIdSId−δS	
dIdt=βWISI+βWId1−βTIdSId−βTIdIId−α+δI	
dIddt=βWIdβTIdSId+βWISTI+βWIdSTId+βTIdIId−α+δId	
dSTdt=βTId1−βWIdSId−βWISTI−βWIdSTId−α+δST	
Parameters	
b	Natural birth rate in the population	
I	Number of infected individuals with only the wild type of HIV	
Id	Individuals infected with both HIV and TIPs	
ST	Individuals infected with only TIPs	
βWI	Transmission rate of wild type HIV from HIV infected individuals	
βWId	Transmission rate of wild type HIV from dually infected individuals	
βTId	Transmission rate of TIPs from dually infected individuals	

Coupling within-host and between-host scales of HIV immunoepidemiological models

The potential for transmission between HIV+ individuals to susceptibles is affected by the viral load of infected hosts (Attia et al., 2009). In all the models that we analyzed in this systematic review, the transmission rate between hosts is dependent on the within-host viral load. We categorize the models into those where the transmission rate is a function of viral load and those where the equilibria of the within-host model are used to determine the transmission rate.

HIV transmission rate as a function of viral load

The within-host and between-host scales of HIV immunoepidemiological models are coupled by basing the transmission rate on the time-varying viral load since infection. The viral load (and thus the transmission rate) is high during the acute and late stages of HIV infection while being low during the latent stage (Hollingsworth, Anderson & Fraser, 2008; DebRoy & Martcheva, 2008). Table 10 shows the formulation of this model. Unlike the basic SI epidemiological model that assumes constant transmission rate (β), the between-host model assigns time-varying transmission rate, which is dependent on the non-linear viral immune dynamics of HIV in the within-host model.

Table 10 Coupling mechanism of within-host and between-host scales of HIV dynamics.

The within-host and between-host layers of HIV multi-scale model are linked using partial differential equations. The HIV viral immune dynamics model (see Table 2) determines the time-varying within-host viral load, which impacts the transmission rate (β(τ) = r.V(τ); r is a constant coefficient). Another method to determine the HIV transmission rate is based on the viral load equilibrium.

Equations	
dSdt=b−S∫0∞βτIτ,tdτ−δS	
∂I∂t+∂I∂τ=−mVτIτ,t	
I0,t=S∫0∞βτIτ,tdτ	
Parameters	
S	Number of individuals in the susceptible class	
I(τ, t)	Number of infected individuals structured by time since infection (τ)	
b	Natural birth rate in the population	
β(τ)	HIV transmission rate (r.V(τ))	
m	Coefficient on dependence of induced mortality due to disease on the host viral load.	

In some models, the transmission rate depends on the viral load continuously over time (Shen, Xiao & Rong, 2015; Martcheva & Li, 2013; Saenz & Bonhoeffer, 2013). Saenz and Bonhoeffer also distinguished between drug-resistant and drug-sensitive strains and their corresponding impact on the transmission rate (Saenz & Bonhoeffer, 2013). Martcheva and Li made the death of infected individuals depend on the viral load over time, since the AIDS stage is associated with high viral load (Martcheva & Li, 2013).

In the context of HIV evolution, while the transmission rate varies through time depending on the viral load, the viral load is also modeled to distinguish between different strains (Doekes, Fraser & Lythgoe, 2017; Lythgoe, Pellis & Fraser, 2013). The transmission rate depends on a predefined infectivity profile which changes depending on the stage of infection, and the frequency of the different viral strains in an infected population. Doekes et al. made the transmission rate depend on the frequency of viral strains that were only in actively infected CD4+ T cells (Doekes, Fraser & Lythgoe, 2017).

The within-host viral load can be used to individualize the transmission rate over time (Yeghiazarian, Cumberland & Yang, 2013; Sun et al., 2016). The CD4+ T cell count can also be used to determine the stage of infection (Yeghiazarian, Cumberland & Yang, 2013).

HIV transmission rate using viral load equilibrium

Another method of linking the within-host and between-host scales is to use the within-host model to determine an equilibrium for the viral load. This equilibrium can then be used as a constant parameter in the between-host model, which can then be analyzed further by differing the parameters of the within-host model (Metzger, Lloyd-Smith & Weinberger, 2011; Cuadros & García-Ramos, 2012). Cuadros & García-Ramos (2012) accounted for the amplified viral load due to co-infection and the corresponding increase in HIV transmission rate. Metzger, Lloyd-Smith & Weinberger (2011) determined the differing viral loads associated with HIV and TIPs, and their effect on the transmission probabilities between infected populations.

Clinical and public health implications

HIV virulence

Clinical studies have shown that HIV has evolved an intermediate level of virulence at the within-host level that optimizes the transmission potential of the virus at the population level (Fraser et al., 2007). However, at the within-host level, HIV can evolve quickly (Lemey, Rambaut & Pybus, 2006), virulence increases during the course of the infection (Kouyos et al., 2011), and infections with higher replicative capacities have higher virulence (Kouyos et al., 2011). Replicative capacities also increase over the course of infection, albeit slowly (Kouyos et al., 2011). Because of this behavior of HIV at the within-host level, it might be expected that HIV would evolve a high virulence at the within-host level, even if it did not optimize the transmission potential at the population level. To understand these seemingly contradictory results, immunoepidemiological models were used, which incorporated these behaviors of HIV at the within-host level (Lythgoe, Pellis & Fraser, 2013; Doekes, Fraser & Lythgoe, 2017). The model developed by Lythgoe, Pellis & Fraser (2013) found that small rates of within-host evolution optimize the transmission potential at the population level, whereas higher rates of within-host evolution lead to high levels of virulence, but lower transmission potential. Lythgoe, Pellis & Fraser (2013) suggest that the clinical observations seen in HIV may be a result of a within-host fitness landscape that is complex to traverse, since this leads to smaller rates of within-host evolution. They also suggest the effect of the adaptive immune response may play a role in explaining the observed behavior (Lythgoe, Pellis & Fraser, 2013). Based off the results from Lythgoe et al., a similar model was constructed by Doekes, Fraser & Lythgoe (2017), which included a latent reservoir of CD4+ T cells at the within-host level. They found that this latent reservoir may be responsible for delaying the evolutionary dynamics at the within-host level, which then leads to the transmission potential being optimized (Doekes, Fraser & Lythgoe, 2017).

Antiretroviral therapy

While there is uncertainty over the timing of initiating antiretroviral therapy, some studies have suggested there may be benefits to beginning treatment early (When To Start Consortium et al., 2009; Cohen, 2011). Experimental studies also suggest that because ART reduces transmissibility, increasing coverage levels may reduce the prevalence of HIV (Cohen, 2011). However, drug-resistant strains can emerge, which can lead to treatment failure (Hué et al., 2009; Hamers et al., 2011). Immunoepidemiological models were used to understand these effects of ART, focusing on treatment timing (Sun et al., 2016; Yeghiazarian, Cumberland & Yang, 2013), coverage levels (Shen, Xiao & Rong, 2015), and drug resistance (Saenz & Bonhoeffer, 2013; Sun et al., 2016). The models showed that, in general, initiating treatment early (Yeghiazarian, Cumberland & Yang, 2013; Sun et al., 2016), increasing coverage (Shen, Xiao & Rong, 2015; Saenz & Bonhoeffer, 2013), and increasing effectiveness of ART (Shen, Xiao & Rong, 2015; Saenz & Bonhoeffer, 2013) reduces the prevalence of HIV.

However, in certain cases, increases in the prevalence of HIV may occur even with early treatment initiation, increased coverage, and increased effectiveness of ART to drug-sensitive strains. Models showed that as ART coverage levels increase, the prevalence of drug-resistant strains increase, which cause an increase in HIV prevalence (Saenz & Bonhoeffer, 2013). Prevalence can also increase if drug-resistant strains cause the drug efficacy to decrease significantly (Sun et al., 2016). These results imply that there may be an optimal therapy coverage level that will minimize the number of infections (Saenz & Bonhoeffer, 2013). Therefore, in these cases, the models suggest that HIV prevalence can be reduced by focusing efforts on decreasing the risk of drug resistance emergence (Saenz & Bonhoeffer, 2013).

Clinical studies have observed that under certain conditions, the prevalence of HIV increases when ART coverage levels increase (Zaidi et al., 2013). Zaidi et al. (2013) hypothesize that since ART reduces viral load, patients may live longer, and thus have the ability to infect more people. Immunoepidemiolgical models also observed this effect (Shen, Xiao & Rong, 2015), including a model of super-infection (Martcheva & Li, 2013). Both model outcomes are consistent with the hypothesis of Zaidi et al., since the models find that the increased prevalence is due solely to decreases in viral load (Shen, Xiao & Rong, 2015; Martcheva & Li, 2013). The model developed by Shen, Xiao & Rong (2015) found that this effect can be minimized if drug effectiveness is high.

Therapeutic interfering particles

Clinical trials have shown that therapeutic interfering particles (TIPs) have the potential to reduce within-host viral load (Levine et al., 2006) and transmit between hosts (Aaskov et al., 2006). Experimental studies have also shown that HIV transmission rates between hosts depend on the within-host viral load (Fraser et al., 2007). Based on these assumptions, an immunoepidemiological model is developed, which deploys TIPs to a small proportion (1%) of the population (Metzger, Lloyd-Smith & Weinberger, 2011). The effect on HIV prevalence due to deploying TIPs is compared to deploying ART and to deploying a hypothetical HIV vaccine. When TIPs have the ability to transmit between hosts, the model shows deploying TIPs reduces HIV prevalence to lower levels than deploying ART therapy or deploying vaccines. However, the model shows that if TIPs do not have the ability to transmit between hosts, then there is minimal effect on the reduction of HIV prevalence (Metzger, Lloyd-Smith & Weinberger, 2011). While more study of TIPs is needed, TIPs have the potential to be an effective therapy than either ART or vaccines.

HIV co-infection

Experimental studies suggest that co-infection may be responsible for increases seen in set-point viral load (spVL) at the within-host level over time (Modjarrad & Vermund, 2010). These increases due to co-infection vary substantially within-host (Kublin et al., 2005). Also, the concentrations of co-infection in high-risk groups versus low-risk groups may affect how HIV spreads in the general population (Abu-Raddad et al., 2008). To study the mechanisms responsible for these effects of co-infection, an immunoepidemiological model was developed (Cuadros & García-Ramos, 2012). They found that populations with higher spVL lead to higher increases in viral load due to co-infection, whereas populations with lower spVL leads to lower increases in viral load due to co-infection. This leads to a greater chance of co-infection increasing the prevalence of HIV in populations with high average spVL (Cuadros & García-Ramos, 2012). Therefore, the effects of co-infection may be mitigated by identifying the viral factors that can reduce the spVL in the population.

Discussion

Mathematical implementation of HIV immunoepidemiological models

We conducted this systematic review of HIV immunoepidemiological models to improve our understanding and analysis of the synergistic dynamics of HIV prognoses at the individual level and the transmission dynamics at the population level. With respect to mathematical implementation, within-host models are implemented using ordinary differential equations which determine the HIV transmission rate for the between-host model. If the within-host model is used at equilibrium to determine constant parameters for the between-host model, ordinary differential equations are used for the between-host model as well (Cuadros & García-Ramos, 2012; Metzger, Lloyd-Smith & Weinberger, 2011). Integro-differential equations with delay are used in the between-host scales of HIV immunoepidemiological models to study HIV evolution dynamics (Lythgoe, Pellis & Fraser, 2013; Doekes, Fraser & Lythgoe, 2017). Partial differential equations are used for the between-host model if the transmission rate changes continuously with the within-host viral load over time (Shen, Xiao & Rong, 2015; Martcheva & Li, 2013; Saenz & Bonhoeffer, 2013). Individual or agent-based based models analyze the HIV transmission dynamics between individual agents in a population, wherein the HIV transmission rates of each individual is determined by their specific within-host immune-viral dynamics (Sun et al., 2016; Yeghiazarian, Cumberland & Yang, 2013).

Complexity of multi-scale models

Multi-scale HIV immunoepidemiological models have higher complexity in comparison to single-scale immune or epidemiology models (Mideo, Alizon & Day, 2008). Thereby, the choice of immunoepidemiological models should be determined by problems with significant public health and clinical implications that can be addressed better by multi-scale models compared to single-scale models.

Table 11 Clinical and public health relevant problems of HIV dynamics.

Clinical and public health relevant problems of HIV dynamics that can be potentially addressed using multi-scale models.

• How does the time-varying viral load and shedding rate since HIV infection impact the transmission rate between hosts?	
• How does co-infection among HIV-infected individuals impact the HIV dynamics in the population?	
• How does super-infection of multiple HIV strains among infected individuals impact the HIV dynamics in the population?	
• How does within-host mutations of drug-sensitive and drug-resistant strains impact the HIV evolution in the population?	
• How does timing of treatment initiation among infected individuals impact the HIV dynamics in the population?	
• How does treatment compliance and interruption behavior of HIV-positive individuals impact HIV dynamics in the population?	
• What is the impact of pre-exposure prophylaxis of high-risk HIV-negative individuals on HIV dynamics in the population?	
• How can multi-scale HIV models be verified and validated with empirical data?	
• How can the optimal layers from micro-biological (genomic, molecular, cellular, organ) to macro-social (individual, family, community, national, global) levels for multi-scale models of HIV dynamics be selected appropriately?	

Clinical and public health relevant problems of HIV dynamics

Table 11 illustrates the clinical and public health relevant problems of HIV virulence, co-infection, super infection, drug resistance and treatment dynamics that can be potentially addressed using multi-scale models. Since the viral load among infected individuals varies with time during the acute, latent and late stages of HIV infection, immunoepidemiological models account for the time-varying viral load within host and their impact on transmission between hosts. Co-infection among HIV-infected individuals increases the average set-point of viral load in the population (Cuadros & García-Ramos, 2012). Super-infection of multiple HIV strains leads to oscillations in the population level which do not occur in the absence of super-infection; this effect is only observed using the multi-scale immunoepidemiological model (Martcheva & Li, 2013). The emergence of drug resistance within hosts impacts the optimal coverage levels of drug-sensitive treatment at the population level (Saenz & Bonhoeffer, 2013). Immunoepidemiological models can account for treatment initiation, compliance and interruption behavior among HIV-positive individuals as well as pre-exposure prophylaxis of high-risk HIV-negative individuals, and their impact on emergence of drug resistance in the population. The new knowledge gained from analysis of HIV immunoepidemiological dynamics add value in improving clinical and public health interventions for prevention and control of HIV epidemics.

Limitations

We reviewed English language articles on HIV immunoepidemiological models that were referenced in the PubMed database. The dynamics of the HIV immunoepidemiological models are dependent on the selection of parameters, and the coupling mechanisms of within-host immune-viral dynamics and between-host transmission dynamics. Verification and validation of HIV immunoepidemiological models (and multi-scale models in general) with empirical data is a challenge to be addressed in future studies. Also, the selection of optimal layers from the genomic, molecular, cellular, and organ levels at the micro-biological scale to the individual, family, community, national, and global levels at the macro-social scale is a challenge that need be addressed well in future studies.

Conclusion

HIV immunoepidemiological models combine the immune-viral dynamics at the within-host immunological scale with the transmission dynamics at the between-host epidemiological scale to analyze HIV dynamics of a single strain infection, co-infection, super-infection, evolution, drug resistance, and treatment protocols in heterogeneous populations. Based on our understanding of synergistic dynamics of HIV at the individual and population scales, we should select the optimal layers of analysis from micro-biological to macro-social levels for multi-scale models to identify and improve solutions to clinical and public health relevant problems of HIV dynamics.

Supplemental Information

Supplemental Information 1 PRISMA flow diagram

Click here for additional data file.

Supplemental Information 2 PRISMA checklist

Click here for additional data file.

Supplemental Information 3 Rationale and contribution of systematic review

Click here for additional data file.

Additional Information and Declarations

Competing Interests

Author Contributions

Data Availability

The authors declare there are no competing interests.

Nargesalsadat Dorratoltaj and Ryan Nikin-Beers conceived and designed the experiments, performed the experiments, analyzed the data, contributed reagents/materials/analysis tools, wrote the paper, prepared figures and/or tables, reviewed drafts of the paper.

Stanca M. Ciupe and Stephen G. Eubank contributed reagents/materials/analysis tools, reviewed drafts of the paper.

Kaja M. Abbas conceived and designed the experiments, performed the experiments, analyzed the data, contributed reagents/materials/analysis tools, wrote the paper, reviewed drafts of the paper.

The following information was supplied regarding data availability:

This paper is a systematic review. Thereby, we did not generate, collect or analyze any raw data or code. Nine papers included the systematic review, and references to these nine papers are included in the paper (see Table 1).

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
