# Peer review of "Multi-scale immunoepidemiological modeling of within-host and between-host HIV dynamics: systematic review of mathematical models"

_PeerJ, doi:10.7717/peerj.3877_

## Round 0.1 · original submission · Major Revisions

Dear authors,

The three reviewers have indicated several concerns about your paper, which you should addressed before publication. Therefore, my decision is MAJOR REVISION.

With respect and warm regards,
Dr Palazón-Bru (academic editor for PeerJ)

Reviewer 1 ·

Basic reporting

Clearly written. Manuscript's structure is appropriate.

Experimental design

Goal and justification of this study, as a systematic review, are clearly stated. Methodology of systematic review is well presented.

Validity of the findings

Conclusions are well stated.

Additional comments

1. Assuming viral shedding as part of the basic viral dynamics model (line128, page 7; Table 2) is uncommon. A more representative example is found in Perelson 2002, Modelling viral and immune system dynamics, Nature Reviews.
2. Do not use "et al." when there are only two authors; e.g. "Martcheva and Li" instead of "Martcheva et al" (line 134, page 8)—also in lines 140,153, 181, 220, 222, 238.
3. Adding a formulation of model's summary for within host scale models for "HIV evolution" (lines144-149) and "HIV and therapeutic interfering particles" (lines 158-164) would improve the introduction of models (similar to Tables 2, 3, etc.).
4. Note that Saenz & Bonhoeffer (2013) also consider acute, latent and late stages of HIV infection, so this should be mentioned in the corresponding section (lines 177-185).
5. In section "HIV transmission rate as a function of viral load" (lines 212-233), mention the different types of functions that are used (e.g., linear, Hill's).
6. A better reference for the sentence "The viral load (and thus the transmission rate) is high during the acute and late stages of HIV infection while being low during the latent stage" (lines 214-216) is: Hollingsworth et al. 2008, HIV-1 transmission by stage of infection, Journal of infectious diseases.
7. A few typos in References list (e.g., "hiv" in line 376).
8. Models diagrams (Tables 2, 3, 4, 5, 6, 7) should include state variables when stating rates, otherwise they are inconsistent (e.g. "k" is not a rate in the same sense as "d").
9. The reproduction rate of T related to Tm is missing in the model diagram of Table 5.
10. Note that not all studies considered in the manuscript used the function r*V(t) as implied in Table 8.

Reviewer 2 ·

Basic reporting

'no comment'

Experimental design

'no comment'

Validity of the findings

'no comment'

Additional comments

The manuscript is well organized. Thank authors for writing the manuscript carefully.

1. The manuscript is a review article. The review was done in a systematic way as described in lines 102 and in Figure 2.
2. The authors reviewed 9 articles selected from 89 found in the PubMed database. The inclusion and exclusion criteria were clearly defined in the METHODS section (line #98).
3. The authors summarized the results found in those selected papers and stated in the RESULTS section starting at line # 109. One of the key findings is the within host viral load that enhances between host infection. They also highlighted the impact of super infection, antiretroviral therapy, drug resistance, treatment at early or late stages on HIV infection and prevalence.
4. The authors though saw the increased complexity of the multi-scale modeling, but found significant public health impact of these models.
5. This review should be helpful for further studies to eliminate the long time persisted disease HIV-AIDS.

With regards to your concern “regarding the assessment of the experimental design or the validity of the findings were provided”- The authors of the manuscript did not conduct any experiments. So, there was no issue of validation of findings. However, they have followed a systematic review process that I mentioned in my opinion (1,2) above.

Based on my findings I have no objection to accept this manuscript for publication.

Reviewer 3 ·

Basic reporting

The authors are making a systematic review of the use of immunoepidemiological mathematical models of HIV. Overall, they found 9 articles where such models were used. Although the paper contains interesting pieces of information, the basic reporting would require modifications. A clear definition of the objectives is likely the most important issue, followed by a need for synthesis and better cohesion between the elements of this paper.
Without a clear understanding of the review’s objectives, it was difficult to assess both the relevance and adequacy of the results. I assumed that the authors meant to review how, in the mathematical models, the within-host evolution of HIV was considered to affect the transmission of the disease at the population level. To that regard, it was difficult to find this information in the paper. In terms of cohesion, working on the link between the pieces of information that are reported would help, as the sentences sometimes gave the impression to be “floating”, like bullet points. A clearer definition of the objectives would greatly help in that regard, as the information could be linked back to these objectives throughout the paper and hence give a better sense of direction to the reader. The authors could also consider discarding some of the information that is given, depending on these objectives.

In terms of the context, the authors should consider elaborating on the clinical relevance of immunoepidemiological models. Maybe a good way would be to describre models that consider, separately, the within-host or the between-host HIV dynamics, especially the questions they were able to answer but mostly those that cannot be answered because they are not considered altogether. Finding more precise and clinical research questions relative to what is written in lines 77-80 could help improve motivation. Also, is it the first time immunoepidemiological models are reviewed in the context of HIV?

As for the Results section, is it possible that some sentences from the Results were statement of facts or data-driven conclusions (e.g. lines 133-134, 159-160, 178-180, 187, 191-192), rather than model elements of the reviewed paper? If this is the case, these sentences would be a better fit in the Introduction. I would avoid dividing the results in very short sections and work on synthesizing the information. Some statements were ambiguous, e.g. sentences like “… can be modeled similar to co-infection (Metzger et al. 2011)… (line 161)” left me to wonder if the referenced authors are the ones stating that it could be modeled, or that they actually included this element in their model and, if so, if they were the only one considering TIPs. I believe important that the authors identify these sentences and make more precise statements. I suggest systematically reporting the frequency of papers having considered which specific model element and why they have done so. Maybe a table with one line per model and check marks for elements considered (columns) could help?

Table 1 contained the most interesting information. I suggest elaborating more on the important aspects of it. Consider discarding Figure 1, Tables 2 to 9, which did not bring much to the reading, unless they can be related to an objective. Table 9 could help with motivation (see above comment), but I would also add why it is important to assess these questions.

Experimental design

The general methodology seems adequate for a systematic review. However, I was wondering if the authors considered using MeSH terms, from PubMED; this is a great tool to find all terms related to concepts. Also, I suggest looking at other databases, as mathematical models can be underrepresented in PubMED.

Validity of the findings

I was unable to relate the discussion to the objectives, for the issues stated above.

Additional comments

No further comment.

---

## Round 0.2 · Major Revisions

Dear authors,

After reading the report of Reviewer 3, I think your manuscript still needs some major changes before publication. Therefore, my decision is TO REVISE.

Warm regards,
Dr Palazón-Bru (academic editor for PeerJ)

Reviewer 1 ·

Basic reporting

No comment

Experimental design

No comment

Validity of the findings

No comment

Additional comments

All my previous concerns/comments have been satisfactorily answered. I have no further comments.

Reviewer 3 ·

Basic reporting

The authors made modifications which improved my understanding of the nature of the reported results. Some of my concerns remain. My main concern pertained to the clarity of the stated objectives (see below for details). To this end, I understand that no modification was made by the authors. Consequently, my troubles relating the reported results to the objectives remain. Of relatively lesser importance is the text’s lack of direction. A lot of information is given and it would help a lot if the authors explained, before the Results section, how they chose which information to report and the purpose of reporting this information. I also had a bit of difficulties finding the pieces of information which I thought most relevant to the paper.

I leave it to the editor to decide if the latter issues are important enough to be tackled.

Objective
To be more precise, I struggled understanding the last part of this sentence: “The objective of this study is to conduct a systematic review of multi-scale HIV immunoepidemiological models to infer the synergistic dynamics of HIV prognoses at the individual level and the transmission dynamics at the population level.” Is this what the authors meant: “The objective of this study is to conduct a systematic review of multi-scale HIV immunoepidemiological models to better understand how the evolution of the virus within the hosts could impact the transmission of the disease between individuals, and vice-versa.” ?

Please clarify in the text.

Results
Most of the Results section focuses on technical aspects of the modeling approach. What is the general purpose of presenting this information? Please provide a rationale in the text. In my mind and if I understand the objectives, the authors wished to inform the synergy between within- and between-host dynamics. If this is the case, then model inferences can inform this synergy (Table 1), not the modeling approach. The modeling approach, on the other hand, informs the reader of the validity of model inferences (see suggestion below for details). Although it is quite possible that this was the authors’ intent in presenting the modeling approach, how the results are presented does not help the reader make the appropriate assessment of inferences.

Accordingly, here are some modification suggestions:

- Most models have a great number of underlying assumptions (e.g. homogeneous populations, deterministic/stochastic nature of a relationship, etc.). Please mention why the specific model assumptions were chosen to be reported in the objectives. As an example, you could write: “Inferences from mathematical models are sensitive to model assumptions. To enable a better assessment of the validity of inferences, we will report the model assumptions that we believe had a larger impact on model predictions.”

- In the Results section, it would be advisable not to separate data-driven facts, model assumptions and model inferences, and make it clear which is which. As an example based on lines 275-279 (this is not meant to be accurate!): “There is evidence suggesting that deploying therapeutic interfering particles in even a small proportion of infected individuals reduces the prevalence of HIV to low levels. (ref=longitudinal study) This is thought to be due to TIPs’ ability to transmit between hosts and and target high-risk groups. (ref=source of this opinion) In one immunoepidemiological model, this was modelled by assuming that the susceptible population infected by TIP carriers become TIP carriers themselves, and those have one-half the transmission rate of a non-TIP carrier.(ref=model paper) This assumption was supported by clinical data. Results from analyzing this model suggest that TIP may reduce the challenges of ART therapy and vaccines, and can be complementary to both.(ref=model paper) This is due to [particular aspect of the model dynamics] and adds to the current knowledge ...”

- If I understood your objectives correctly, I suggest describing the most important information relative to your objectives in the text rather than in a table (e.g. Table 1). I would put less relevant technical aspects of the modeling approach in Appendix.

Experimental design

no comment

Validity of the findings

no comment

---

## Author Rebuttal · Round 0.2

June 26, 2017

Dr. Palazón-Bru
Editor
Dear Dr. Palazón-Bru,

Thank you for the opportunity to revise our manuscript, "Multi-scale Immunoepidemiological Modeling of Within-host and Between-host HIV Dynamics: Systematic Review of Mathematical Models."

We are grateful to you and the reviewers for their valuable comments and feedback. We have updated the manuscript to address all the comments.

We are submitting the revised manuscript, and thank you for your consideration for publication in PeerJ.

Kind regards,
Kaja
(On behalf of all the authors)

Kaja Abbas
Assistant Professor of Infectious Diseases in Public Health
Department of Population Health Sciences
Virginia Tech

**Response to feedback of Reviewer #1**

We thank Reviewer #1 for your valuable feedback. We have updated the manuscript to address your comments, as illustrated below.

**Comment:** Basic reporting

Clearly written. Manuscript's structure is appropriate.

**Response:**

Thank you for your kind feedback.

**Comment:** Experimental design

Goal and justification of this study, as a systematic review, are clearly stated. Methodology of systematic review is well presented.

**Response:**

Thank you for your kind feedback.

**Comment:** Validity of the findings

Conclusions are well stated.

**Response:**

Thank you for your kind feedback.

**Comment R1.1:**

Assuming viral shedding as part of the basic viral dynamics model (line128, page 7; Table 2) is uncommon. A more representative example is found in Perelson 2002, Modelling viral and immune system dynamics, Nature Reviews.

**Response:**

While including viral shedding is not common in a basic viral dynamics model, we have included it due to its biological relevance in the immunoepidemiological dynamics. Table 2 presents the within-host layer of a multi-scale HIV immunoepidemiological model, and the viral shedding rate has a negative effect on the viral load within-host. The viral load (which is time-varying) impacts the HIV transmission rate in the within-host layer of the multi-scale HIV immunoepidemiological model.

**Comment R1.2:**

Do not use "et al." when there are only two authors; e.g. "Martcheva and Li" instead of "Martcheva et al" (line 134, page 8)—also in lines 140,153, 181, 220, 222, 238.

**Response:**

We have updated the manuscript to include both authors instead of "et al".

**Comment R1.3:**

Adding a formulation of model's summary for within host scale models for "HIV evolution" (lines144-149) and "HIV and therapeutic interfering particles" (lines 158-164) would improve the introduction of models (similar to Tables 2, 3, etc.).

**Response:**

We have added two new tables: Table 8 for HIV evolution and Table 9 for HIV and therapeutic interfering particles. The schematic diagrams and corresponding equations have been added to the two tables. In Table 8 for HIV evolution, the diagram and its corresponding functions show HIV transmission dynamics between infected individuals with different strains. Infected individuals with strains i may get infected with another strain j and transmit the dominant strain of HIV. In Table 9 for HIV and therapeutic interfering particles (TIP), the diagram and corresponding functions show HIV transmission dynamics between infected individuals with wild type of HIV and TIPs. Individuals can get infected with one type or both. Individuals who are infected with wild type of HIV or TIPs can get reinfected with both types.

**Comment R1.4:**

Note that Saenz & Bonhoeffer (2013) also consider acute, latent and late stages of HIV infection, so this should be mentioned in the corresponding section (lines 177-185).

**Response:**

We have updated the first paragraph of the section "Acute, latent and late stages of HIV infection" (line 177) by adding the reference to Saenz & Bonhoeffer (2013) as follows:

The infected population is categorized depending on the stage of infection, and the transmission rates differ depending on whether the infected population is in the acute, latent, or AIDS stages (Cuadros & Garcia-Ramos (2012); Yeghiazarian et al. (2013); Sun et al. (2016); Shen et al. (2015); Saenz & Bonhoeffer (2013)).

**Comment R1.5:**

In section "HIV transmission rate as a function of viral load" (lines 212-233), mention the different types of functions that are used (e.g., linear, Hill's).

**Response:**

We have added the following sentence in the first paragraph of the section "HIV transmission rate as a function of viral load" (line 212), to clarify that the HIV transmission rate is dependent on the non-linear viral immune dynamics of HIV in the within-host model.

"Unlike the basic *SI* epidemiological model that assumes constant transmission rate ($\beta$), the between-host model assigns time-varying transmission rate, which is dependent on the non-linear viral immune dynamics of HIV in the within-host model."

**Comment R1.6:**

A better reference for the sentence "The viral load (and thus the transmission rate) is high during the acute and late stages of HIV infection while being low during the latent stage" (lines 214-216) is: Hollingsworth et al. 2008, HIV-1 transmission by stage of infection, Journal of infectious diseases.

**Response:**

We have added the Hollingsworth et al. 2008 reference to the sentence, and is updated as follows:

The viral load (and thus the transmission rate) is high during the acute and late stages of HIV infection while being low during the latent stage (Hollingsworth et al. (2008); DebRoy & Martcheva (2008)).

**Comment R1.7:**

A few typos in References list (e.g., "hiv" in line 376).

**Response:**

We have updated the references list and taken care of the typos.

**Comment R1.8:**

Models diagrams (Tables 2, 3, 4, 5, 6, 7) should include state variables when stating rates,

otherwise they are inconsistent (e.g. "k" is not a rate in the same sense as "d").

**Response:** The manuscript has been updated with the modified diagrams, and includes the

state variables while stating rates.

**Comment R1.9:**

The reproduction rate of T related to Tm is missing in the model diagram of Table 5.

**Response:**

We have updated Table 5, and the model diagram includes the reproduction rate of T related to

$T_m$.

**Comment R1.10:**

Note that not all studies considered in the manuscript used the function r*V(t) as implied in

Table 8.

**Response:**

We have added the following sentence to the legend description of Table 8 (Table 10 in the new

revised paper) to clarify that the HIV transmission rate can be determined in

immunoepidemiological models using the time-varying viral load or based on the viral load equilibrium.

"Another method to determine the HIV transmission rate is based on the viral load equilibrium."

**Response to feedback of Reviewer #2**

We thank Reviewer #2 for your valuable feedback. We have updated the manuscript to address your comments, as illustrated below.

**Comment:**

The manuscript is well organized. Thank authors for writing the manuscript carefully.

**Response:**

Thank you for your kind feedback.

**Comment R2.1:**

The manuscript is a review article. The review was done in a systematic way as described in lines 102 and in Figure 2.

**Response:**

Yes, this manuscript is a systematic review, and we followed the PRISMA (Preferred Reporting Items for Systematic Reviews and Meta-Analyses) framework in conducting this study.

**Comment R2.2:**

The authors reviewed 9 articles selected from 89 found in the PubMed database. The inclusion and exclusion criteria were clearly defined in the METHODS section (line #98).

**Response:**

Yes, 9 articles were found eligible for this systematic review.

**Comment R2.3:**

The authors summarized the results found in those selected papers and stated in the RESULTS section starting at line # 109. One of the key findings is the within host viral load that enhances between host infection. They also highlighted the impact of super infection, antiretroviral therapy, drug resistance, treatment at early or late stages on HIV infection and prevalence.

**Response:**

Yes, based on the HIV immunoepidemiological models that we analyzed in this systematic review, the transmission rate between hosts is dependent on the within-host viral load. We categorized the models into those where the transmission rate is a function of viral load and those where the equilibria of the within-host model are used to determine the transmission rate. The models are useful to analyze the within-host dynamics of HIV super-infection, co-infection, drug resistance, evolution, and treatment in HIV+ individuals, and their between-host impact on the epidemic pathways in the population.

**Comment R2.4:**

The authors though saw the increased complexity of the multi-scale modeling, but found significant public health impact of these models.

**Response:**

Yes, while the multi-scale HIV immunoepidemiological models add complexity, they are useful to address clinical and public health relevant problems of HIV dynamics, as illustrated in Table 11.

**Comment R2.5:**

This review should be helpful for further studies to eliminate the long time persisted disease HIV-AIDS.

**Response:**

We conducted this systematic review of HIV immunoepidemiological models to improve our understanding and analysis of the synergistic dynamics of HIV prognoses at the individual level and the transmission dynamics at the population level. Yes, we expect that this review will be useful in developing future HIV immunoepidemiological modeling studies on clinical and public health relevant problems of HIV dynamics.

**Comment:**

With regards to your concern "regarding the assessment of the experimental design or the validity of the findings were provided"- The authors of the manuscript did not conduct any experiments. So, there was no issue of validation of findings. However, they have followed a systematic review process that I mentioned in my opinion (1,2) above.

**Response:**

Yes, we conducted a systematic review by following the PRISMA (Preferred Reporting Items for Systematic Reviews and Meta-Analyses) framework.

**Comment:**

Based on my findings I have no objection to accept this manuscript for publication.

**Response:**

Thank you for your valuable and kind feedback.

**Response to feedback of Reviewer #3**

We thank Reviewer #3 for your valuable feedback. We have updated the manuscript to address your comments, as illustrated below.

**Basic reporting**

**Comment:**

The authors are making a systematic review of the use of immunoepidemiological mathematical models of HIV. Overall, they found 9 articles where such models were used. Although the paper contains interesting pieces of information, the basic reporting would require modifications. A clear definition of the objectives is likely the most important issue, followed by a need for synthesis and better cohesion between the elements of this paper.

Without a clear understanding of the review's objectives, it was difficult to assess both the relevance and adequacy of the results. I assumed that the authors meant to review how, in the mathematical models, the within-host evolution of HIV was considered to affect the transmission of the disease at the population level. To that regard, it was difficult to find this information in the paper. In terms of cohesion, working on the link between the pieces of information that are reported would help, as the sentences sometimes gave the impression to be "floating", like bullet points. A clearer definition of the objectives would greatly help in that regard, as the information could be linked back to these objectives throughout the paper and hence give a better sense of direction to the reader. The authors could also consider discarding some of the information that is given, depending on these objectives.

**Response:**

The objective of this systematic review is to infer the synergistic and coupled dynamics of HIV prognoses at the individual level and transmission dynamics at the population level. HIV

immunoepidemiological models broadly focus on the following questions (as mentioned in the "Clinical and public health significance" subsection of Introduction section:

- How does within-host immune-viral dynamics of HIV affect incidence at the population level?

- How does population level transmission dynamics of HIV affect viral evolution at the individual level?

As discussed in the "Clinical and public health relevant problems of HIV dynamics" subsection of Discussion section and in Table 11 -- By conducting this systematic review, we have synthesized specific clinical and public health problems of HIV virulence, co-infection, super infection, drug resistance and treatment dynamics that can be addressed through HIV immunoepidemiological models.

**Comment:**

In terms of the context, the authors should consider elaborating on the clinical relevance of immunoepidemiological models. Maybe a good way would be to describe models that consider, separately, the within-host or the between-host HIV dynamics, especially the questions they were able to answer but mostly those that cannot be answered because they are not considered altogether. Finding more precise and clinical research questions relative to what is written in lines 77-80 could help improve motivation.

**Response:**

This manuscript focuses on the coupled impact of immunological and epidemiological dynamics of the HIV infection within-host and HIV transmission between-host. The broad questions that can be answered through multi-scale model of HIV dynamics are mentioned in "Clinical and public health significance" subsection under Introduction section, and specific questions of

clinical and public health relevance that we synthesized through the systematic review are illustrated in the "Clinical and public health problems of HIV dynamics" subsection under Discussion section and in Table 11.

**Comment:**

Also, is it the first time immunoepidemiological models are reviewed in the context of HIV?

**Response:**

Based on our knowledge and literature search, this study is the first systematic review of mathematical models of HIV immunoepidemiological models.

**Comment:**

As for the Results section, is it possible that some sentences from the Results were statement of facts or data-driven conclusions (e.g. lines 133-134, 159-160, 178-180, 187, 191-192), rather than model elements of the reviewed paper? If this is the case, these sentences would be a better fit in the Introduction. I would avoid dividing the results in very short sections and work on synthesizing the information. Some statements were ambiguous, e.g. sentences like "… can be modeled similar to co-infection (Metzger et al. 2011)… (line 161)" left me to wonder if the referenced authors are the ones stating that it could be modeled, or that they actually included this element in their model and, if so, if they were the only one considering TIPs. I believe important that the authors identify these sentences and make more precise statements. I suggest systematically reporting the frequency of papers having considered which specific model element and why they have done so. Maybe a table with one line per model and check marks for elements considered (columns) could help?

**Response:**

On lines 133-134, we edited the paragraph to make it clearer that the statement about the "strain with the larger reproduction rate becomes dominant" is a modeling assumption based on competitive exclusion at the within-host scale.

"Martcheva and Li included HIV infection with multiple strains in their model, with the assumption of complete competitive exclusion between the strains at the within-host scale. In this context, the strain with the larger reproduction rate becomes dominant. They studied the impact of virulence of different strains on the equilibrium at the individual and population scales (Martcheva & Li (2013))."

On lines 159-161, we modified the paragraph by eliminating the reference to "similar to co-infection." We stated that HIV and TIPs are treated as separate viral strains near the beginning of the description of the model.

"Therapeutic interfering particles (TIPs) are an emerging drug therapy where therapeutic versions of the pathogen are manufactured to attack viral replication processes and can be transmitted between hosts (Metzger et al. (2011)). In the within-host model developed by Metzger et al, HIV and TIPs are treated as separate viral strains. The model includes CD4+ T cells infected with HIV only, CD4+ T cells infected with TIPs only, and CD4+ T cells dually infected with HIV and TIPs (Metzger et al. (2011))."

On lines 178-180, we made clear that transmission rates differ depending on stage on infection is a data-driven conclusion by Hollingsworth et al (2008). These models then incorporate this conclusion into their models.

"Previous studies have shown that transmission rates differ depending on whether the infected population is in the acute, latent, or AIDS stages (Hollingsworth et al. 2008). This conclusion can be incorporated into immunoepidemiological models by categorizing the infected population into different stages (Cuadros & Garcıa-Ramos (2012); Yeghiazarian et al. (2013); Sun et al. (2016); Shen et al. (2015); Saenz & Bonhoeffer (2013))."

In line 187, we changed the paragraph to clarify that the statement "Depending on the within-host dominant strain among the infected individuals, susceptibles are at risk of infection with the dominant strain" is a modeling assumption.

"Due to the assumption of competitive exclusion at the within-host level in the model developed by Martcheva and Li, susceptible individuals only become infected with one of the strains. Thus, only infected individuals having the dominant within-host strain can super-infect individuals with the lesser within-host strain (Martcheva & Li (2013))."

In lines 191-192, we clarify the context of when drug-resistant strains can emerge and why Saenz and Bonhoeffer put these categories into their model. We modify the paragraph to include the following.

"Drug-resistant strains can emerge during ART (Rong et al. 2007), or can be transmitted between individuals who have never been exposed to ART (Hue et al. 2009), which may lead to treatment failure if ART is begun (Hamers et al. 2011). Saenz & Bonhoeffer thus categorize the infected population into those with only drug-sensitive or only drug-resistant strains with or without treatment, and those with drug-sensitive strains that develop drug-resistance while receiving treatment (Saenz & Bonhoeffer (2013))."

We divide the models into short sections so that they can be easily referenced. We synthesize the main results of each model into Table 1. We also incorporate the results into the Discussion section.

Table 1 separates the studies by topics they investigated. The previous sentences which we have modified also explain any model assumptions.

**Comment:**

Table 1 contained the most interesting information. I suggest elaborating more on the important aspects of it.

**Response:**

The objective, model implementation, immunoepidemiological link between within-host and between-host models, and significant inferences of these studies are summarized in Table 1. We believe that these are the most important aspects of these studies.

**Comment:**

Consider discarding Figure 1, Tables 2 to 9, which did not bring much to the reading, unless they can be related to an objective.

**Response:**

The other 2 reviewers have commented that the manuscript is well organized. Reviewer #1 has requested two additional tables for summarizing the HIV immunoepidemiological models for "HIV evolution" and "HIV and therapeutic interfering particles", and we have added these two tables as well. We believe that Figure 1 and Tables 2 to 11 provide valuable information for

readers new to research on immunoepidemiological models in general, and in particular on HIV immunoepidemiological models.

**Comment:**

Table 9 could help with motivation (see above comment), but I would also add why it is important to assess these questions.

**Response:**

We have added the following sentence to the end of the subsection "Clinical and public health problems of HIV dynamics" subsection under Discussion section:

The new knowledge gained from analysis of HIV immunoepidemiological dynamics add value in improving clinical and public health interventions for prevention and control of HIV epidemics.

**Experimental design**

**Comment:**

The general methodology seems adequate for a systematic review. However, I was wondering if the authors considered using MeSH terms, from PubMED; this is a great tool to find all terms related to concepts.

**Response:**

We did not use MeSH terms in our search. Searching with MeSH will capture results indexed with MeSH but may miss anything not yet indexed, while keywords will capture anything not indexed with MeSH terms.

**Comment:**

Also, I suggest looking at other databases, as mathematical models can be underrepresented in PubMED.

**Response:**

We have included this issue as a limitation that we only reviewed English language articles on HIV immunoepidemiological models that were referenced in only the PubMed database.

**Validity of the findings**

**Comment:**

I was unable to relate the discussion to the objectives, for the issues stated above.

**Response:**

The objective of this systematic review is to infer the synergistic and coupled dynamics of HIV prognoses at the individual level and transmission dynamics at the population level. HIV immunoepidemiological models broadly focus on the following questions (as mentioned in the "Clinical and public health significance" subsection of Introduction section:

- How does within-host immune-viral dynamics of HIV affect incidence at the population level?

- How does population level transmission dynamics of HIV affect viral evolution at the individual level?

As discussed in the "Clinical and public health relevant problems of HIV dynamics" subsection of Discussion section and in Table 9 -- By conducting this systematic review, we have synthesized specific clinical and public health problems of HIV virulence, co-infection, super infection, drug resistance and treatment dynamics that can be addressed through HIV immunoepidemiological models.

---

## Round 0.3 · accepted · Accept

Dear authors,

The manuscript has high standards to be published in PeerJ in its current form. Therefore, my decision is TO ACCEPT.

Congratulations!


With respect and warm regards,
Dr Palazón-Bru (academic editor for PeerJ)